# Combined Treatment with Sodium-Glucose Cotransporter-2 Inhibitor (Canagliflozin) and Dipeptidyl Peptidase-4 Inhibitor (Teneligliptin) Alleviates NASH Progression in A Non-Diabetic Rat Model of Steatohepatitis

**DOI:** 10.3390/ijms21062164

**Published:** 2020-03-21

**Authors:** Takahiro Ozutsumi, Tadashi Namisaki, Naotaka Shimozato, Kosuke Kaji, Yuki Tsuji, Daisuke Kaya, Yukihisa Fujinaga, Masanori Furukawa, Keisuke Nakanishi, Shinya Sato, Yasuhiko Sawada, Soichiro Saikawa, Koh Kitagawa, Hiroaki Takaya, Hideto Kawaratani, Mitsuteru Kitade, Kei Moriya, Ryuichi Noguchi, Takemi Akahane, Akira Mitoro, Hitoshi Yoshiji

**Affiliations:** Department of Gastroenterology, Nara Medical University, 840 Shijo-cho, Kashihara, Nara 634-8522, Japan; ozutaka@naramed-u.ac.jp (T.O.); shimozato@naramed-u.ac.jp (N.S.); kajik@naramed-u.ac.jp (K.K.); tsujih@naramed-u.ac.jp (Y.T.); kayad@naramed-u.ac.jp (D.K.); fujinaga@naramed-u.ac.jp (Y.F.); furukawa@naramed-u.ac.jp (M.F.); nakanishi@naramed-u.ac.jp (K.N.); shinyasato@naramed-u.ac.jp (S.S.); yasuhiko@naramed-u.ac.jp (Y.S.); saikawa@naramed-u.ac.jp (S.S.); Kitagawa@naramed-u.ac.jp (K.K.); htky@naramed-u.ac.jp (H.T.); kawara@naramed-u.ac.jp (H.K.); Kitadem@naramed-u.ac.jp (M.K.); moriyak@naramed-u.ac.jp (K.M.); rnoguchi@naramed-u.ac.jp (R.N.); stakemi@naramed-u.ac.jp (T.A.); mitoroak@naramed-u.ac.jp (A.M.); yoshijih@naramed-u.ac.jp (H.Y.)

**Keywords:** non-alcoholic steatohepatitis, hepatic fibrogenesis, hepatocarcinogenesis, canagliflozin, teneligliptin

## Abstract

Hepatocellular carcinoma (HCC) is the strongest independent predictor of mortality in non-alcoholic steatohepatitis (NASH)-related cirrhosis. The effects and mechanisms of combination of sodium-dependent glucose cotransporter inhibitor and canagliflozin (CA) and dipeptidyl peptidase-4 inhibitor and teneligliptin (TE) on non-diabetic NASH progression were examined. CA and TE suppressed choline-deficient, L-amino acid-defined diet-induced hepatic fibrogenesis and carcinogenesis. CA alone or with TE significantly decreased proinflammatory cytokine expression. CA and TE significantly attenuated hepatic lipid peroxidation. In vitro studies showed that TE alone or with CA inhibited cell proliferation and TGF-β1 and α1 (I)-procollagen mRNA expression in Ac-HSCs. CA+TE inhibited liver fibrogenesis by attenuating hepatic lipid peroxidation and inflammation and by inhibiting Ac-HSC proliferation with concomitant attenuation of hepatic lipid peroxidation. Moreover, CA+TE suppressed in vivo angiogenesis and oxidative DNA damage. CA or CA+TE inhibited HCC cells and human umbilical vein endothelial cell (HUVEC) proliferation. CA+TE suppressed vascular endothelial growth factor expression and promoted increased E-cadherin expression in HUVECs. CA+TE potentially exerts synergistic effects on hepatocarcinogenesis prevention by suppressing HCC cell proliferation and angiogenesis and concomitantly reducing oxidative stress and by inhibiting angiogenesis with attenuation of oxidative stress. CA+TE showed chemopreventive effects on NASH progression compared with single agent in non-diabetic rat model of NASH, concurrent with Ac-HSC and HCC cell proliferation, angiogenesis oxidative stress, and inflammation. Both agents are widely, safely used in clinical practice; combined treatment may represent a potential strategy against NASH.

## 1. Introduction

Non-alcoholic fatty liver disease (NAFLD) includes the entire spectrum of fatty liver disease, ranging from simple steatosis to non-alcoholic steatohepatitis (NASH), significant liver fibrosis, cirrhosis, and hepatocellular carcinoma (HCC). We have shown that blockade of angiotensin-II (AT-II) signaling at the AT-II type-1 receptor served to inhibit hepatic fibrogenesis together with the suppression of hepatic stellate cell (Ac-HSC) activation [1]. Blockade of the renin-angiotensin system with clinically relevant doses of angiotensin (AT)1R blocker (ARB: losartan) was found to inhibit both hepatocarcinogenesis [2] and progression of HCC [3] while at the same time inhibiting vascular endothelial growth factor (VEGF)-mediated neovascularization. It is important to recognize that the relationship between activated HSCs and the pathogenesis of NASH has been considered with respect to the multiple parallel hits hypothesis, and includes factors such as endotoxin and oxidative stress that are currently perceived as key contributors to the progression from fatty liver to NASH [4]. We have also recently shown that administration of fructose resulted in an increase in the extent of transport of endogenous gut-derived bacterial endotoxin by the portal vein; this leads to liver fibrosis and hepatocarcinogenesis via induction of lipopolysaccharide (LPS)/Toll-like receptor 4 signaling in a choline-deficient, l-amino-acid-defined (CDAA)-fed rat model [5]. Miura et al. have shown that proinflammatory cytokines and receptors including the C-C motif chemokine receptor 2 (CCR2), tumor necrosis factor-α (TNFα) and interleukin 6 (IL-6) derived from Kupffer cells all contribute to the progression of NASH and to the development of HCC in the CDAA diet-induced NASH model [6,7,8,9].

NASH is frequently accompanied by metabolic disorders such as type 2 diabetes mellitus (T2DM) [10]. We previously reported that the SGLT2-I, ipragliflozin, successfully ameliorates liver fibrosis in diabetic rats by improving insulin sensitivity [11]. The anti-cancer effects of canagliflozin (CA) have been attributed to inhibition of mitochondrial complex I and mitochondrial respiration [12]. Furthermore, SGLT2 is functionally expressed in various cancers including pancreatic, prostate [13], liver [14], and colon cancers [15], and SGLT2 inhibitors (SGLT2-Is) exhibit antitumor effects on these cancers [16]. We have recently shown that SGLT2 inhibitors may function as novel anti-cancer agents by inhibiting angiogenesis and progression to HCC in a mouse xenograft mouse model [17]. Dipeptidyl peptidase-4 inhibitors (DPP4-Is) are commonly prescribed for the treatment of T2DM. Two randomized controlled trials recently showed that use of the DPP4-I, sitagliptin, alone does not significantly attenuate liver fibrosis and liver fat in patients with NAFLD [18,19]. DPP4 has been identified as a multifunctional glycoprotein involved in different biological processes, including inflammation, liver fibrosis, malignant transformation, and tumor immunity [20,21]. It is highly expressed in the liver, especially in the endothelial cells (ECs), and its expression is increased in HCC [22]. While we observed that sitagliptin monotherapy exerted significant inhibitory effects on liver fibrogenesis and carcinogenesis in a rat model of steatohepatitis [23], it may be difficult to completely suppress the cumulative development of liver fibrosis and HCC using a single agent in clinical practice [2,24]. Our previous study has shown that the combined use of sitagliptin and either losartan or obeticholic acid showed a synergistic repressive effect on NASH development in experimental models [22,25]. Japan has just recently approved the use of a fixed-dose combined tablet of teneligliptin (20 mg) and canagliflozin (100 mg). As such, the impact of clinically equivalent doses of CA (100 mg/day) and teneligliptin (TE) (20 mg/day) [26] and their impact on the progression of NASH progression was the focus of this study. Furthermore, the combined use of empagliflozin (SGLT2-I) and linagliptin (DPP4-I) exhibited antifibrotic effects accompanied by antisteatotic and anti-inflammatory effects in diabetic mice [27]. The present study examined the effects of CA and TE on hepatic fibrogenesis and carcinogenesis, as well as the potential underlying mechanisms in a rat model of NASH.

## 2. Results

### 2.1. General Findings

The baseline characteristics of the six-week-old F344 rats are showed in Table 1. The final body weights of the choline-sufficient, L-amino acid-defined (CSAA)-fed rats (G1) were greater than those that of the CDAA-fed rats (G2–G5). The relative liver weights of the G2–G5 rats were greater than those of the G1 rats. Serum alanine aminotransferase (ALT) levels were significantly augmented in the G2–G5 rats compared with the G1 rats and were significantly lower in G3 and G5 compared with G2. There were no significant differences in serum concentrations of albumin, total bilirubin, and triglyceride among the groups. Plasma glucagon, serum glucose, and insulin levels, and the quantitative insulin sensitivity check index were not significantly altered in G3 or G4 rats. Similarly, no significant differences in hepatic glucagon-like peptide-1 (GLP-1) mRNA expression levels were observed among the experimental groups (data not shown).

### 2.2. Effects of Canagliflozin (CA) and Teneligliptin (TE) on Hepatic Fibrogenesis

Extensive collagen deposition in liver was observed in the G2 rats. Hepatic fibrogenesis was significantly reduced in G3 and G4 compared with that of G2 rats. CA+TE (G5) showed a more potent inhibitory effect than either monotherapy (Figure 1). No fibrosis development was observed in G1. Immunohistochemistry analysis revealed a significantly reduced number of α-smooth muscle actin (SMA)-positive Ac-HSCs after treatment with CA and TE (Figure 2a). A significant decrease in α-SMA-positive Ac-HSC numbers was observed following treatment with CA and TE. Computer-assisted semiquantitative analysis of α-SMA immunohistochemistry revealed a decrease in the α-SMA staining area as well as suppression of hepatic fibrogenesis (Figure 2b). CA and TE treatment significantly inhibited hepatic mRNA expression of transforming growth factor (TGF-β1) and α1(I)-procollagen compared with that in the CDAA diet group (G2; Figure 3a,b). Similar to the effect on liver fibrosis, CA+TE showed more potent inhibitory effects on hepatic expression of TGF-β1 and α1(I)-procollagen than did the effects of either single agent. The inhibitory effects of CA and TE were of a comparable magnitude to the observed effects on liver fibrosis.

### 2.3. In vitro Effects of CA and TE on Ac-HSCs

No significant microscopic morphologic changes were observed in HSCs among the five groups during the experimental period. Treatment with TE, either alone or combined with CA, significantly attenuated Ac-HSC proliferation (Figure 4a) as well as TGF-β1 (Figure 4b) and α1(I)-procollagen (Figure 4c) mRNA expression. On the other hand, treatment with CA alone showed no significant effect on either Ac-HSC proliferation or TGF-β1 and α1(I)-procollagen mRNA expression.

(A) Ac-HSC proliferation was attenuated by single treatment using CA and TE, whereas CA+TE showed a stronger suppressive effect compared with both monotherapies. CA and TE (G5) showed significantly stronger inhibition of TGF-β1 (B) and α1(I)-procollagen (C) mRNA expression in Ac-HSC compared with the single treatment groups (G3, G4). These inhibitory effects closely matched the changes observed in Ac-HSC proliferation.

### 2.4. Effects of CA and TE on Hepatic Inflammatory Cytokine Levels

Levels of TNF-α, IL-6, and CCL2 in the liver significantly increased in G2 rats (fed a CDAA diet) compared with those in G1. However, the CDAA diet-induced-increases in these three parameters were significantly decreased in G3 and G5 (treated with CA and CA + TE, respectively), whereas treatment with TE alone did not significantly influence these parameters (Figure 5).

Levels of hepatic (A) TNF-α, (B) IL-6, and (C) CCL2 were significantly increased in the choline-deficient, L-amino acid-defined (CDAA) diet group (G2) compared with the CSAA diet group (G1). The CDAA diet-induced increase in these three parameters was significantly decreased in CA-treated (G3) and CA + TE (G5) groups. However, no significant reduction was observed in these parameters in TE-treated group (G4). Values represent mean ± SD (*n* = 10). **p* < 0.05, ** *p* < 0.01.

### 2.5. Effects of CA and TE on Preneoplastic Lesion Development

The effects of CA and TE on placental glutathione S-transferase (GST-P)-positive preneoplastic lesions in conjunction with neovascularization were examined. The livers of rats on the CDAA diet for 16 weeks exhibit features of fatty cirrhosis, including numerous and various sized neoplastic nodules with aberrant histologic, architectural, and cytoplasmic features together with nuclear atypia (Figure 6a,b). Semiquantitative analysis showed a significant decrease in both GST-P-positive preneoplastic lesion size and number in G2 rats compared with those in G1. Similar to the effects on liver fibrogenesis, combined treatment with CA and TE showed stronger inhibitory effects compared with use of either single agent (Figure 6c). No GST-P-positive lesions were observed in G1 rats. CA+TE exerted more potent suppressive effects on hepatic CD31 and VEGF mRNA expression compared with use of either single agent (Figure 7a,b). These effects closely matched the suppression of GST-P-positive preneoplastic lesions.

### 2.6. Effects of CA and TE on Hepatocellular Carcinoma (HCC) Cells and Endothelial Cells (ECs) in vitro

Treatment using CA alone or in combination with TE significantly inhibited proliferation of two human HCC cell lines (HepG2 and Huh7) (Figure 8a,b) and ECs (Figure 8c). However, TE showed no significant effect on the proliferation of these three cell types. We performed a more extensive examination of the in vitro effects of CA and TE on the expression of E-cadherin and VEGF in primary human umbilical vein endothelial cell (HUVEC) culture. CA and TE, each acting alone, promoted increased expression of the epithelial marker, E-cadherin, and at the same time suppressed the expression of the angiogenesis marker, VEGF, in HUVEC culture (Figure 8d,e). Simultaneous administration of both agents resulted in more potent stimulatory and inibitory effects on E-cadherin and VEGF expression, respectively, than detected in response to either agent alone.

### 2.7. Effects of CA and TE on Hepatic Oxidative Stress

Reactive oxygen species (ROS) contribute to NASH progression; therefore, we examined the effects of CA and TE on markers of lipid peroxidation and oxidative DNA damage, particularly MDA and 8-OHdG, respectively. Rats fed a CDAA diet showed elevated levels of malondialdehyde (MDA) and 8-hydroxy-2'-deoxyguanosine (8-OHdG) in the liver (Figure 9a,b), and these levels were significantly decreased in G3 and G4 compared with G2. The suppressive effects of CA and TE on MDA and 8-OHdG were almost equivalent, and CA+TE exerted a more potent inhibitory effect than either single agent.

### 2.8. Changes in the Non-Alcoholic Fatty Liver Disease (NAFLD) Activity Score

Microscopic examination revealed significant reductions in steatosis, lobular inflammation and hepatocellular ballooning in groups G3 and G5 (Figure 10 and Table 2) (i.e., those treated with CA and CA + TE, respectively) compared to what was observed among the G2 rats (fed a CDAA diet). These changes were accompanied by a significant decrease in the alanine aminotransferase (ALT) level, indicating that CA, but not TE, had both cytoprotective and anti-inflammatory effects on target hepatocytes.

### 2.9. Measurement of Serum TE Concentration

Serum TE concentration was 1.73 ± 0.27 and 1.88 ± 0.31 nmol/L in G4 and G5, respectively, which were higher than the IC_50_ value of TE (1.14 nmol/L).

### 2.10. Effects of CA and TE on DPP4 Activity

DPP4 activity was significantly augmented in G2 compared with in G1 and was significantly suppressed in G4 and G5 compared with G2 (Figure 11).

## 3. Discussion

The present study found that CA and TE markedly attenuated CDAA diet-induced hepatic fibrogenesis and carcinogenesis. CA+TE showed greater antifibrotic and anticarcinogenic effects than monotherapy. These inhibitory effects were concurrent with the inhibition of Ac-HSC and HCC cell proliferation, VEGF-mediated neovascularization, ROS, and inflammation. Although monotherapy is preferred [23], complete inhibition of the cumulative development of liver fibrosis and carcinogenesis using a single pharmocotherapeutic agent may be challenging in experiments [24,28,29] and in clinical practice [30]. Therefore, CA and TE may be used as therapeutic agents to target fibrosis in patients with NASH. The substantial therapeutic benefits of CA and TE in slowing NASH progression need to be confirmed for future clinical applications.

CA and TE were effective against liver fibrosis via different action mechanisms, including inhibition of Ac-HSC proliferation, hepatic lipid peroxidation, and inflammation. CA suppressed the activation of protein kinase C [31,32], the NADPH oxidase system, and subsequent ROS production [33,34] in various vascular cells [35]. In addition, we demonstrated that CA potentially inhibits lipid peroxidation in NASH progression. It contributes to inflammatory and fibrotic changes in diabetic kidney disease [36]. Studies investigating the effects of SGLT2-I on chronic inflammation in neural tissues and skeletal muscles have demonstrated that a decrease in the expression of proinflammatory cytokine levels and macrophage accumulation [37] mechanistically underlies the pharmacological activities of canagliflozin. We have previously shown that DPP4, but not SGLT2, expressed by Ac-HSCs and sitagliptin (a DPP4-I) inhibits hepatic fibrosis via the suppression of Ac-HSCs in rats [21]. Increasing evidence from in vitro and in vivo studies suggests that the inhibition of DPP4 attenuates oxidative stress [38,39]. These findings support our theory that CA+TE inhibits liver fibrogenesis by attenuating hepatic lipid peroxidation and inflammation and by inhibiting Ac-HSC proliferation with concomitant attenuation of hepatic lipid peroxidation, respectively.

Consistent with our previous findings [17], CA-mediated inhibition of cell proliferation correlated with G2/M phase arrest and was followed by human HCC cell apoptosis [40]. The concentration of CA (10 µM) used in the present study corresponds to the serum concentrations achieved using clinical doses [41]. CA was shown to promote apoptosis of human HCC cells by enhancing cleavage of caspase [40]. Moreover, the proliferative action of CA occurs in the absence of cell death and is due, in part, to blockade of cyclin A expression inhibition [42]. Conversely, in vitro studies have demonstrated that CA (100 µM) interferes with the anti-proliferative effects on non-small lung cancer cell, indicating that high doses of SGLT2 may inhibit HCC cell proliferation due to cytotoxicity [43]. In contrast, 1–50 µM tofogliflozin (another SGLT2-I) did not modulate the proliferation of SGLT2-expressing human HCC cells [44]. These findings warrant further investigation and future studies should consider the role of different SGLT2-Is to investigate the physiological mechanisms and strategies that protect against NASH progression. These findings suggest that the anti-cancer activity of SGLT2-Is may be at least partially explained by the direct anti-cancer effects of SGLT2-I on HCC cells.

However, the exact mechanisms underlying the effects of CA and TE remain to be elucidated. We recently showed that the inhibitory effects of clinically equivalent doses of DPP4-I on hepatocarcinogenesis were mainly mediated via EC tube formation and oxidative DNA damage and were not due to the direct action of in vitro EC or HCC cell proliferation [22]. The drug, sitagliptin, has recently been shown to have inhibitory effects on NASH-related progression to HCC via its inhibitory actions at the p62/Keap1/Nrf2-pentose phosphate pathway [45]. Furthermore, the dipeptidyl peptidase (DPP)-4 inhibitor, KR62436, promoted primary tumor growth, and lung metastasis via induction of the CXCL12/CXCR4/mTOR/EMT signaling axis in a syngeneic mouse model [46]. The differences in antitumor effects of DPP4 inhibitors might be explained at least in part by differences in pharmacological activities [26]. Conversely, CA was shown to inhibit EC proliferation in a concentration-dependent manner [42]. This suggests that inhibition of both CA and TE synergistically reduces neovascularization. We are currently investigating the direct interaction between CA and TE, angiogenesis, cell proliferation, and ROS during hepatocarcinogenesis. Nevertheless, the present findings indicate that CA and TE potentially exert synergistic effects on the prevention of hepatocarcinogenesis by suppressing HCC proliferation and angiogenesis and concomitantly reducing the oxidative stress and by inhibiting angiogenesis with attenuation of oxidative stress, respectively.

The present study employed a CDAA model to examine the effects of CA and TE on hepatic fibrogenesis and carcinogenesis. The CDAA diet-induced histological changes similar to those observed in human NASH [47]. However, the CDAA dietary model of NASH does not display insulin resistance, obesity, and impaired glucose tolerance, which are clinical characteristics of NASH in humans. The antifibrotic effects of TE were not correlated with local GLP-1 expression in CDAA-induced NASH. The therapeutic impact under conditions of insulin resistance requires further investigation to accurately evaluate the pharmacological effects of drugs.

Collectively, studies have shown that CA and TE have synergistic effects on hepatic fibrogenesis and carcinogenesis with concurrent suppression of HSC and HCC proliferation, angiogenesis, oxidative stress, and inflammation. Furthermore, the inhibitory effects of both agents on NASH progression may be achieved at clinically comparable doses in non-diabetic rats. CA+TE could be more effective in suppressing hepatic fibrosis progression and further preventing fibrosis-associated HCC development in patients with NASH, who are at an increased risk of developing cirrhosis and HCC.

## 4. Materials and Methods

### 4.1. Animals and Reagents

A total of 50 six-week-old male Fischer 344 (F344) rats (CLEA Japan Inc., Tokyo, Japan) were housed in stainless steel mesh cages under the following controlled conditions: temperature, 23 °C ± 3 °C; relative humidity, 50% ± 20%; 10–15 air/h; and 12-h day/night cycle. Animals had ad libitum access to tap water. CA and TE were kindly provided from Mitsubishi Tanabe Pharma Co. Ltd. (Osaka, Japan). Conventional chemical reagents were purchased from Nacalai Tesque (Kyoto, Japan). CDAA and CSAA diets were purchased from CLEA Japan Inc. (Tokyo, Japan).

### 4.2. Animal Treatment

All experiments were performed over a 16-week period. Rats were randomly assigned to five groups: groups 1 to 5 (G1–G5; *n* = 10 per group). Rats in G1 were designated as the negative control group and were fed a CSAA diet, whereas rats in G2–G5 were fed a CDAA diet. Rats in G1 and G2 were administered phosphate-buffered saline via oral gavage once daily during the experimental period. Rats in G3 and G4 were administered clinically equivalent doses of CA (10 mg/kg/day) [17] and TE (0.3 mg/kg/day) [48], respectively, via daily oral gavage. Rats in G5 were treated with a combination of CA and TE. No adverse effects were observed in the present study. There were no differences in food intake among the groups. At end of the study period, rats were anesthetized using isoflurane, and various indices were measured. Serum TE concentration was determined by liquid chromatography–tandem mass spectrometry [49,50]. Analysis was performed using the LCMS-8060 triple quadrupole mass spectrometer with the Nexera UHPLC system (Shimadzu Corporation, Kyoto, Japan) [51]. Serum DPP4 activity was measured using a DPP4 activity assay kit (Biovision, Milpitas, CA, USA) [52]. All procedures were performed according to the criteria outlined in the “Guide for the Care and Use of Laboratory Animals” prepared by the National Academy of Sciences and published by the National Institutes of Health (NIH publication 86–23, revised 1985). All experiments were approved by the Animal Care and Use Committee of Nara Medical University.

### 4.3. Quantitative Real-time Reverse Transcription-Polymerase Chain Reaction (RT-PCR) Analysis

RNA was extracted from powdered frozen liver tissue using the RNeasy Mini Kit (QIAGEN, Tokyo, Japan) and complementary DNA (cDNA) was synthesized from RNA using the High Capacity RNA-to-cDNA kit (Applied Biosystems Inc., Foster City, CA, USA), according to the manufacturer’s instructions. Expression levels of mRNA for TGF-β1 and α1(I)-procollagen were measured in liver tissue and isolated activated hepatic stellate cell (Ac-HSCs), and CD31 and VEGF mRNA was measured in liver tissue and E-cadherin and VEGF mRNA was measured in HUVECs and quantified by RT-PCR using SYBR Green on a Step One Plus sequence-detection system (Applied Biosystems Inc.). PCR was conducted as follows: samples were heated for 20 s at 95 °C followed by 40 cycles of denaturing for 3 s at 95 °C and annealing for 30 s at 60 °C. Glyceraldehyde 3-phosphate dehydrogenase (GAPHD) was used as an endogenous control. The following primer sequences used: TGF-β1, forward 5′-CGGCAGCTGTACATTGACTT-3′ and reverse 5′-AGCGCACGATCATGTTGGAC-3′; α1(I)-procollagen, forward 5′-AGCTCCTGGGCCTATCTGATGA-3′; CD31, forward, 5′-GGCGTCCTGTCCGGAATC-3′ and reverse 5′-AGAACTCCTGCACAGTGACGTATT-3′; VEGF, forward, 5′-CCATGAACTTTCTGCTGTCTT-3′ and reverse 5′-CCATGAACTTTCTGCTGTCTT-3′; E-cadherin forward, 5′-TCCATTTCTTGGTCTACGCC-3′ and reverse 5′-CACCTTCAGCCA ACCTGTTT-3′ and GAPHD, forward 5′-TGGCAAATTCCATGGCA-3′ and reverse 5′-CCTTCTCCATGGTGGT-3′.

### 4.4. Histological and Immunohistochemical Analyses

Liver samples from the five groups were routinely processed to create 5-μm-thick sections of formalin-fixed paraffin-embedded (FFPE) sections for Sirius red staining to evaluate hepatic fibrosis. Immunohistochemistry for α-SMA (DAKO, Kyoto, Japan) and enzyme-altered preneoplastic lesions (GST-P) (MBL Co. Ltd., Nagoya, Japan) was conducted as previously described [53,54]. FFPE tissue sections were semiquantified using ImageJ (National Institutes of Health). Ten microscopic visual fields (×40 magnification) were analyzed per specimen in each group (*n* = 10) using ImageJ analysis, as previously described [55].

### 4.5. Protein Expression Analysis

After determining the protein concentration of 200-mg frozen liver samples, levels of TNF-α (Biosource International, Camarillo, CA), IL-6 (Quantikine HS; R&D Systems, Minneapolis, MN), CCL2 (LifeSpan Biosciences, Seattle, WA, USA), malondialdehyde (MDA, NWLSS, Vancouver, WA, USA), and 8-hydroxydeoxyguanosine (8-OHdG, AbCam, Cambridge, UK) were determined using enzyme-linked immunosorbent assay kits according to the manufacturer’s instructions.

### 4.6. In Vitro Assays

Frozen liver tissue samples were digested in vitro using pre-warmed pronase and collagenase solution. Freshly isolated HSCs from F344 rats were plated at a density of 5 × 10^5^ cells/mL on uncoated plastic dishes [47]. After culture for 5 days, the HSCs showed myofibroblast-like characteristics with reduced lipid vesicles and increased expression of α-SMA. After activation of HSCs by culture on plastic plates for 7 days, all cells were distributed homogeneously and were all α-SMA-positive. HepG2 and Huh7 human liver cancer cells and HUVECs were obtained from the Japanese Cancer Research Resources Bank (Tokyo, Japan). The in vitro effects of CA and TE on the cell proliferation were assessed using a colorimetric assay (Roche Applied Science, Laval, Canada) according to the cleavage of the tetrazolium salt, WST-1, by mitochondrial dehydrogenases to form formazan in viable cells (Roche Applied Science, Laval, Canada). Briefly, Ac-HSCs, HepG2 cells, Huh7 cells, or HUVECs were seeded in 96-well plates with 100 μL of fetal bovine serum-free media. After 48 h incubation, 20 μL of WST-1 reagent (Sigma Aldrich, Poole, UK) was added and plates were incubated for 1 h. Absorbance was measured at 450 nm using a reference wavelength of 620 nm within 30 min using a plate reader.

### 4.7. Statistical Analysis

Data are presented as mean ± standard deviation. One-way analysis of variance followed by Bonferroni or Tukey’s post hoc tests (in vitro study) was performed. All statistical analyses were performed using IBM Statistical Package for the Social Sciences software version 22. All tests were two-tailed and *p*-values of <0.05 were considered significant.

## Figures and Tables

**Figure 1 ijms-21-02164-f001:**
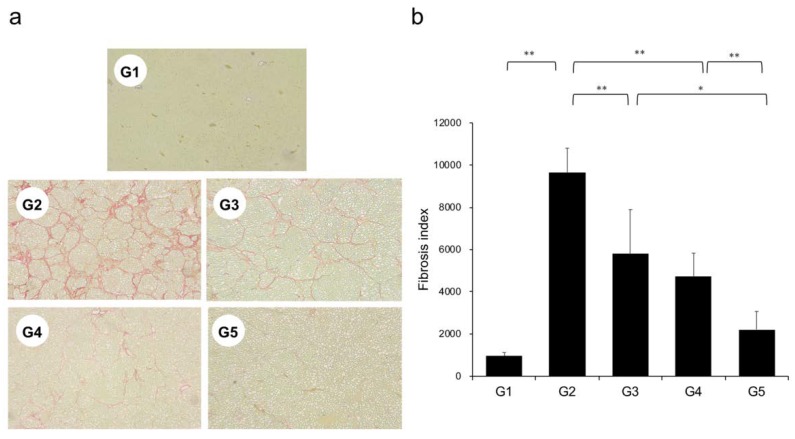
(**a**) Photomicrographs of liver sections stained with Sirius red. (**b**) Collagen content was calculated from Sirius red staining using image analysis software. G1, Group 1 (choline-sufficient, L-amino acid-defined diet); G2, Group 2 (choline-deficient, L-amino acid-defined diet (CDAA)); G3, Group 3 (CDAA+ canagliflozin (CA)); G4, Group 4 (CDAA+ teneligliptin (TE)); G5, Group 5 (CDAA+CA+TE); values represent mean ± SD (*n* = 10). * *p* < 0.05, ** *p* < 0.01.

**Figure 2 ijms-21-02164-f002:**
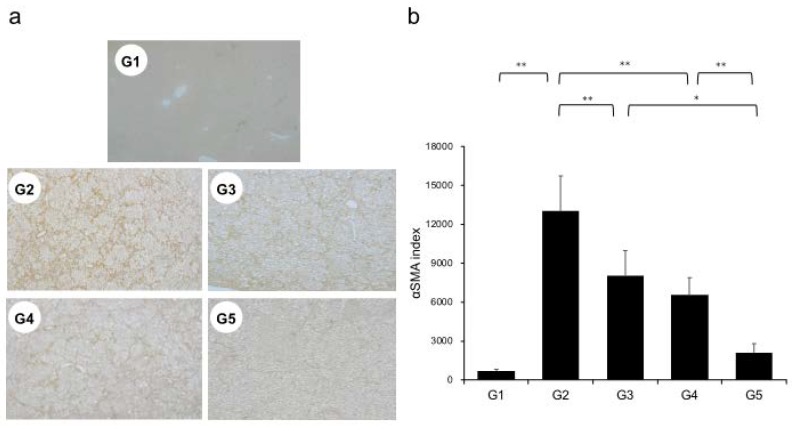
(**a**) Immunohistochemical analysis of alpha-smooth muscle actin (α-SMA) expression. (**b**) α-SMA immunohistochemical staining is assessed using image analysis software. G1, Group 1 (choline-sufficient, L-amino acid-defined diet); G2, Group 2 (choline-deficient, L-amino acid-defined diet (CDAA)); G3, Group 3 (CDAA+ canagliflozin (CA)); G4, Group 4 (CDAA+ teneligliptin (TE)); G5, Group 5 (CDAA+CA+TE); Values represent mean ± SD (*n* = 10). * *p* < 0.05, ** *p* < 0.01.

**Figure 3 ijms-21-02164-f003:**
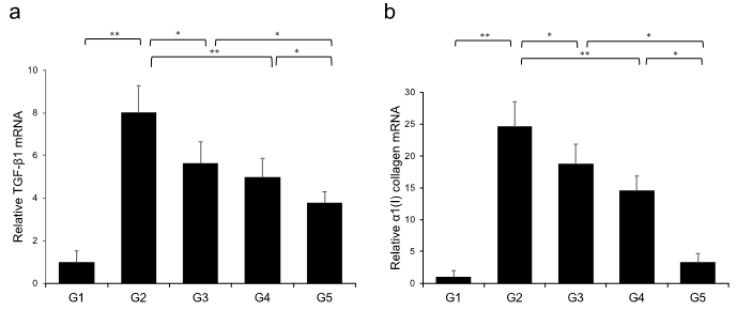
Effects of canagliflozin (CA) and teneligliptin (TE) on hepatic mRNA expression of TGF-β1 (**a**) and α1(I)-procollagen (**b**) in rats fed either the choline-sufficient, L-amino acid-defined (CSAA) diet (G1) or a choline-deficient, L-amino acid-defined (CDAA) diet (G2) and treated with CA (G3), TE (G4), or CA and TE (G5). Data represent mean ± SD (*n* = 10). * *p* < 0.05, ** *p* < 0.01.

**Figure 4 ijms-21-02164-f004:**
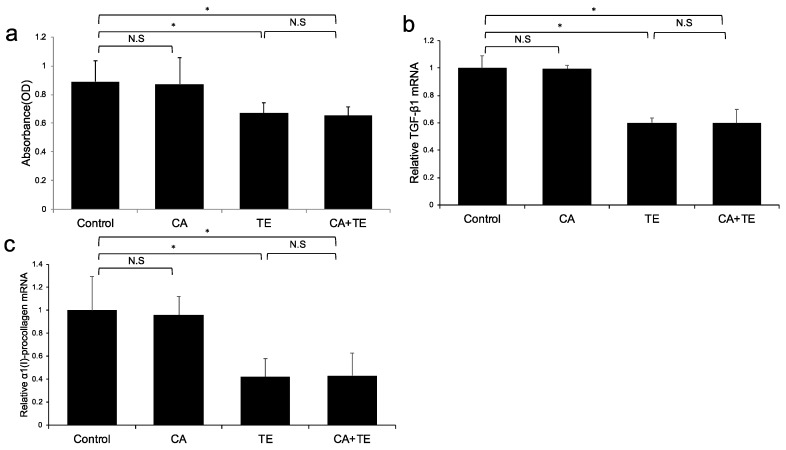
Effects of canagliflozin (CA; 10 μM) and teneligliptin (TE; 5 μM) on cell proliferation (**a**) as well as mRNA expression of TGF-β1 (**b**) and α1 (I)-procollagen (**c**) in activated hepatic stellated cells. The in vitro effects of CA and TE on cell proliferation were assessed using a colorimetric assay. Values represent mean ± SD (*n* = 8). * *p* < 0.05. N.S.: No significance.

**Figure 5 ijms-21-02164-f005:**
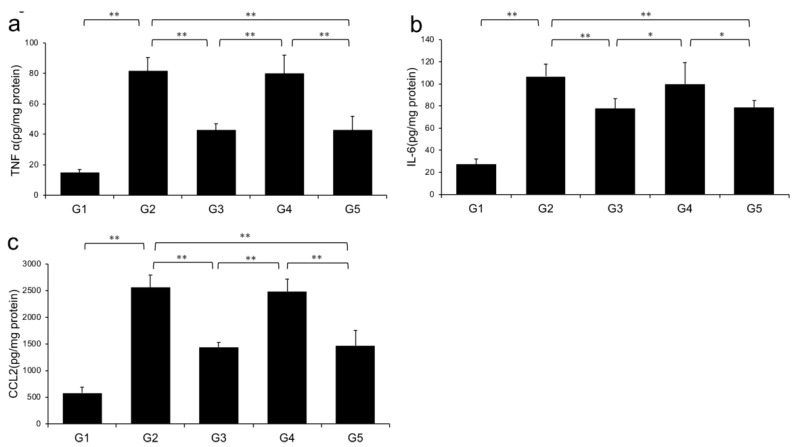
Effects of canagliflozin (CA) and teneligliptin (TE) on hepatic inflammatory cytokines; (**a**) tumor necrosis factor-α (**b**) Interleukin-6 (**c**) C-C motif chemokine 2. G1, Group 1 (choline-sufficient, L-amino acid-defined diet); G2, Group 2 (choline-deficient, L-amino acid-defined diet (CDAA)); G3, Group 3 (CDAA+ canagliflozin (CA)); G4, Group 4 (CDAA+ teneligliptin (TE)); G5, Group 5 (CDAA + CA + TE); Values represent mean ± SD (*n* = 10). * *p* < 0.05, ** *p* < 0.01.

**Figure 6 ijms-21-02164-f006:**
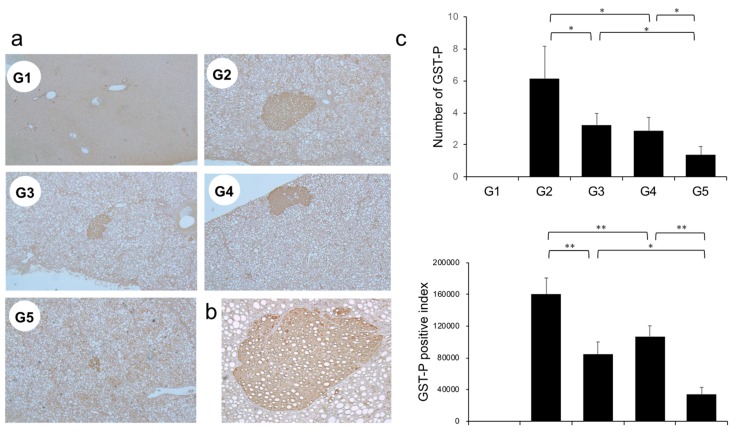
(**a**) Representative photomicrographs of GST-P-positive hepatic preneoplastic lesions. (**b**) Semiquantitative analysis of immunohistochemistry images using ImageJ. (**c**) Higher magnification GST-P-positive hepatic preneoplastic lesions (x100). G1, Group 1 (choline-sufficient, L-amino acid-defined diet); G2, Group 2 (choline-deficient, L-amino acid-defined diet (CDAA)); G3, Group 3 (CDAA + canagliflozin (CA)); G4, Group 4 (CDAA + teneligliptin (TE)); G5, Group 5 (CDAA + CA + TE); Values represent mean ± SD (*n* = 10). * *p* < 0.05, ** *p* < 0.01.

**Figure 7 ijms-21-02164-f007:**
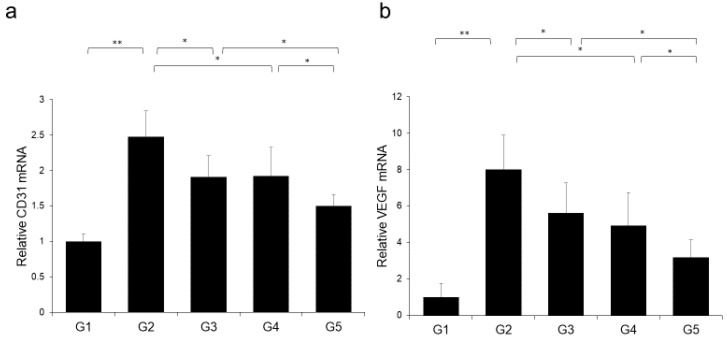
Effects of canagliflozin (CA) and teneligliptin (TE) on hepatic (**a**) CD31 mRNA expression and (**b**) hepatic vascular endothelial growth factor (VEGF) protein expression. G1, Group 1 (choline-sufficient, L-amino acid-defined diet); G2, Group 2 (choline-deficient, L-amino acid-defined diet (CDAA)); G3, Group 3 (CDAA+ canagliflozin (CA)); G4, Group 4 (CDAA+ teneligliptin (TE)); G5, Group 5 (CDAA+CA+TE); Values represent mean ± SD (*n* = 10). * *p* < 0.05, ** *p* < 0.01.

**Figure 8 ijms-21-02164-f008:**
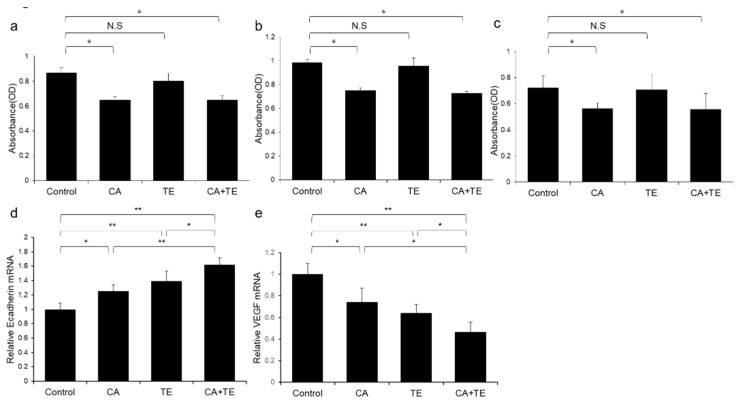
Effects of canagliflozin (CA; 10 μM) and teneligliptin (TE; 5 μM) on cell proliferation of (**a**) HepG2 cells and (**b**) Huh7 cells. Effects of both agents on (**c**) cell proliferation and (**d**) E-cadherin and (**e**) VEGF mRNA expression in human umbilical vein endothelial cells. Values represent mean ± SD (*n* = 8). * *p* < 0.05, ** *p* < 0.01. N.S.: No significance.

**Figure 9 ijms-21-02164-f009:**
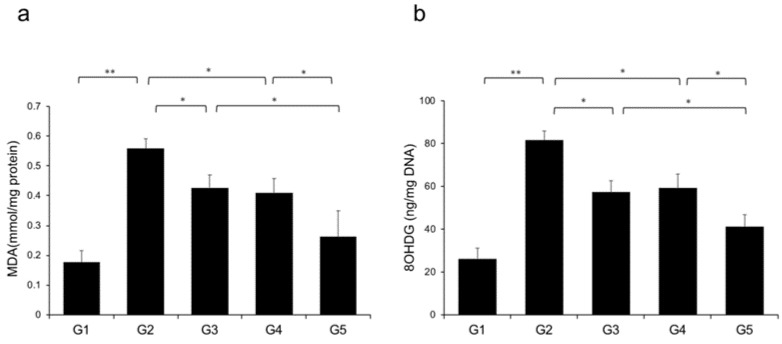
Effects of canagliflozin (CA) and teneligliptin (TE) on hepatic reactive oxygen species (ROS) production; (**a**) malondialdehyde (**b**) 8-hydroxy-2'-deoxyguanosine. G1, Group 1 (choline-sufficient, L-amino acid-defined diet); G2, Group 2 (choline-deficient, L-amino acid-defined diet (CDAA)); G3, Group 3 (CDAA+canagliflozin (CA)); G4, Group 4 (CDAA+ teneligliptin (TE)); G5, Group 5 (CDAA+CA+TE); Values represent mean ± SD (*n* = 10). * *p* < 0.05, ** *p* < 0.01.

**Figure 10 ijms-21-02164-f010:**
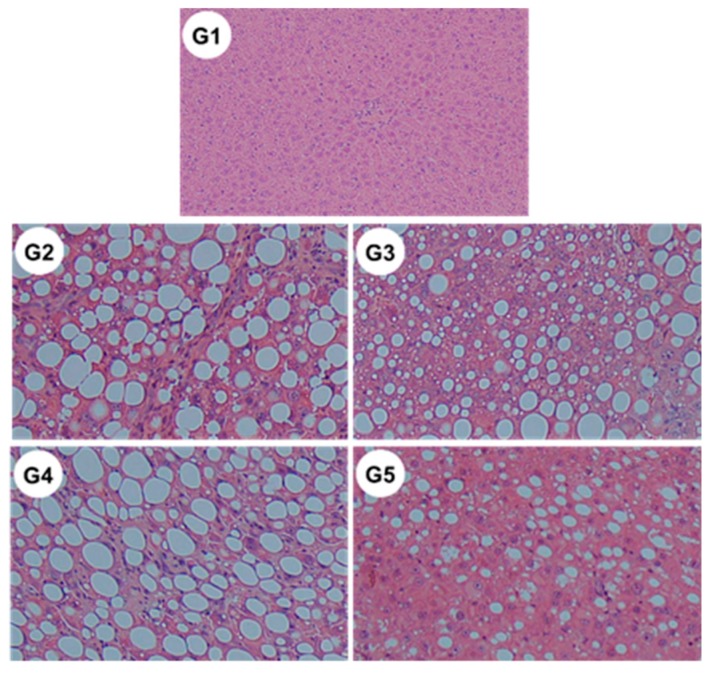
Representative images of hematoxylin–eosin-stained liver sections of different experimental groups (original magnification ×40). G1, Group 1 (choline-sufficient, L-amino acid-defined diet); G2, Group 2 (choline-deficient, L-amino acid-defined diet (CDAA)); G3, Group 3 (CDAA+ canagliflozin (CA)); G4, Group 4 (CDAA+ teneligliptin (TE)); G5, Group 5 (CDAA+CA+TE).

**Figure 11 ijms-21-02164-f011:**
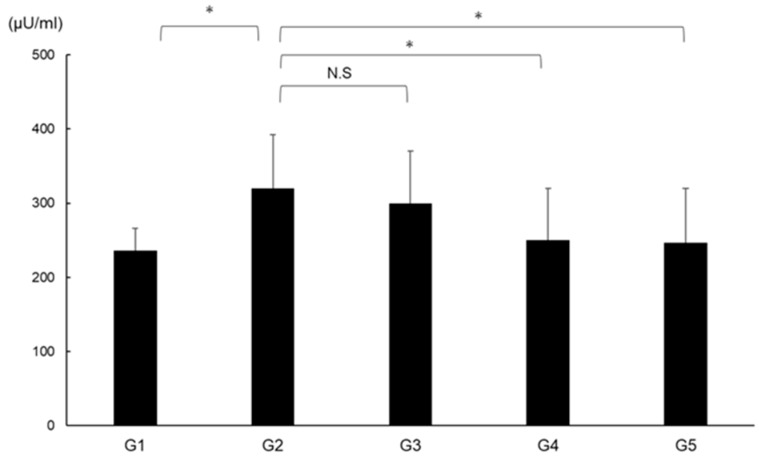
Effects of canagliflozin (CA) and teneligliptin (TE) on serum DPP4 activity. G1, Group 1 (choline-sufficient, L-amino acid-defined diet); G2, Group 2 (choline-deficient, L-amino acid-defined diet (CDAA)); G3, Group 3 (CDAA+ canagliflozin (CA)); G4, Group 4 (CDAA+ teneligliptin (TE)); G5, Group 5 (CDAA + CA + TE). Values represent mean ± SD (*n* = 10). * *p* < 0.05, N.S; not significant.

**Table 1 ijms-21-02164-t001:** Characteristics of the experimental groups in rats fed the CSAA-or CDAA diet.

	CSAA	CDAA	CDAA+CA	CDAA+TE	CDAA+CA+TE
	(G1)	(G2)	(G3)	(G4)	(G5)
Number of rats	10	10	10	10	10
Body weight, g	328.6 ± 15.7	266.1 ± 11.8 ^‡^	258.9 ± 9.6 ^‡^	266.1 ± 14.2 ^‡^	259.3 ± 12.3 ^‡^
Liver/ BW ratio, g/100gBW	3.8 ± 0.1	5.2 ± 0.2 ^‡^	5.4 ± 0.3 ^‡^	5.1 ± 0.5 ^‡^	5.2 ± 0.4 ^‡^
ALT, IU/L	82.2 ± 8.6	297.3 ± 31.0 ^‡^	208.9 ± 13.0 ^‡,†^	278.2 ± 33.1 ^‡^	209.5 ± 29.2 ^‡,†^
ALB, g/dL	3.9 ± 0.1	3.7 ± 0.1	3.8 ± 0.2	3.6 ± 0.3	3.8 ± 0.3
T-Bil, mg/dL	0.13 ± 0.02	0.12 ± 0.03	0.11 ± 0.03	0.13 ± 0.02	0.11 ± 0.03
TG, mg/dL	210.9 ± 39.1	210.4 ± 18.4	208.5 ± 23.2	222.5 ± 37.5	219.3 ± 18.9
Glucose, mg/dL	161.2 ± 36.9	158.4 ± 31.6	138.1 ± 31.5	136.8 ± 21.9	143.4 ± 13.8
Insulin, µU/mL	0.82 ± 0.34	0.86 ± 0.61	0.78 ± 0.40	0.92 ± 0.34	0.89 ± 0.53
Glucagon, pg/mL	30.8 ± 4.8	30.4 ± 6.4	31.2 ± 10.8	29.7 ± 7.3	34.3 ± 12.5
QUICKI	1.53 ± 0.79	1.83 ± 1.1	1.70 ± 0.86	1.29 ± 0.56	1.65 ± 1.01
NEFA, µg/L	466 ± 98.7	440 ± 121.3	401 ± 55.4	386 ± 110.9	430 ± 59.3

Treatment groups; G1, fed choline-sufficient L-amino acid-defined (CSAA) diet; G2, fed choline-deficient L-amino acid-defined (CDAA) diet; G3, CDAA and canagliflozin (10 mg/kg/day); G4, CDAA and teneligliptin (0.3 mg/kg/day); G5, CDAA and canagliflozin(CA) (10 mg/kg/day)+teneligliptin (TE) (0.3 mg/kg/day). Data are represented as mean ± SD. Statistically significant as compared to G1 (^‡^
*p* < 0.01) and G2 († *p* < 0.01). Abbreviations: BW, body weight; ALT, alanine aminotransferase; ALB, albumin; T-Bil, total bilirubin; TG, triglyceride; QUICKI, quantitative insulin sensitivity check index; NEFA, nonesterified fatty acids.

**Table 2 ijms-21-02164-t002:** Non-alcoholic fatty liver disease activity score in experimental groups in a non-diabetic rat model of steatohepatitis.

	CSAA	CDAA	CDAA+CA	CDAA+TE	CDAA+CA+TE
	(G1)	(G2)	(G3)	(G4)	(G5)
**Steatosis**	0	2.8 ± 0.4	2.3 ± 0.5 ^†^	2.5 ± 0.5	1.9 ± 0.5 ^‡^
**Ballooning**	0	1.7 ± 0.5	1.2 ± 0.4 ^†^	1.4 ± 0.5	1.2 ± 0.4 ^†^
**Inflammation**	0	2.4 ± 0.5	1.9 ± 0.3 ^†^	2.3 ± 0.5	1.6 ± 0.7 ^‡^

Treatment groups; G1, fed choline-sufficient L-amino acid-defined (CSAA) diet; G2, fed choline-deficient L-amino acid-defined (CDAA) diet; G3, CDAA and canagliflozin (10 mg/kg/day); G4, CDAA and teneligliptin (10mg/kg/day); G5, CDAA and canagliflozin(CA)(10 mg/kg/day)+teneligliptin (TE) (10 mg/kg/day). Data are represented as mean ± SD. ^†^
*p* < 0.05 compared to CDAA diet,G2. ^‡^
*p* < 0.01 compared to CDAA diet,G2.

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
