# Peer review of "Combined Treatment with Sodium-Glucose Cotransporter-2 Inhibitor (Canagliflozin) and Dipeptidyl Peptidase-4 Inhibitor (Teneligliptin) Alleviates NASH Progression in A Non-Diabetic Rat Model of Steatohepatitis"

_ijms, 2020, doi:10.3390/ijms21062164_

Round 1
Reviewer 1 Report
In this manuscript, authors demonstrate that combined treatment of CA with TE significantly inhibits the progression of CDAA diet-induced hepatic fibrosis. Overall, it would be difficult to assess the novelty of findings in this manuscript and manuscript should be revised.
- In introduction, current description seems less informative. Although CA or TE has been proven as a SGLT2 or a DPP4 inhibitor, respectively, the author did not show any data regarding on its levels or activities in the present study. Thus, authors need to reduce this description part and manifest how NASH is involved in activation of HSC, inflammation, and angiogenesis including key data shown by the authors as well as how important this mechanism should be suppressed by combined therapy. Then, you can highlight these inhibition (SGLT2 or DPP4) as a rational target for combined treatment.
- The description of result section seems to list of a piece of data and has been omitted to describe the rationale between figures. For example, why did the author check the TNF-a, IL6, and CCL2 as pro-inflammatory cytokines? If these cytokines are already known to be specific to CDAA-induced fibrosis, authors need to mention it and put some references. On the other hand, discussion section is too long and less relevant to the main notion.
- In figure legends, just describe the experimental design without interpretation of data. e.g. Figure 4a, it is difficult to know which assay was used for assessing the proliferation and every figure legend included the data interpretation similar to a result section.
Author Response
Reviewer: 1
Reviewer(s)' Comments to Author:
In this manuscript, authors demonstrate that combined treatment of CA with TE significantly inhibits the progression of CDAA diet-induced hepatic fibrosis. Overall, it would be difficult to assess the novelty of findings in this manuscript and manuscript should be revised.
- In the introduction, the current description seems less informative. Although CA and TE are known to inhibit SGLT2 and DPP4, respectively, the author has not shown any data regarding its levels or activities in the present study. Thus, the authors need to shorten the description part and show how NASH is involved in the activation of HSC, inflammation, and angiogenesis, including key data shown by the authors. We recommend that they describe how this important mechanism can be suppressed using combined therapy and highlight these inhibitory actions on SGLT2 and DPP4 as the rational target for combined treatment.
Author response
Thank you for your valuable comments. We have added more information to the manuscript which provides support for our major points. In the revised version, we have discussed findings that indicate that blockade of angiotensin-II (AT-II) signaling at the AT-II type-1 receptor served to inhibit hepatic fibrogenesis together with the suppression of hepatic stellate-cell (Ac-HSC) activation (see new reference No. 1: Yoshiji H et al., Hepatology 2001). Furthermore, blockade of the renin-angiotensin system (RAS) with clinically relevant doses of angiotensin (AT)1R blocker (ARB: losartan) was found to inhibit both hepatocarcinogenesis (new reference No. 2: Yoshiji H et al., J Gastroenterol 2014) and progression of hepatocellular carcinoma (HCC; new reference No. 3: Yoshiji H et al., Curr Cancer Drug Targets 2011) while at the same time inhibiting vascular endothelial growth factor (VEGF)-mediated neovascularization. It is important to recognize that the relationship between activated HSCs and the pathogenesis of NASH has been considered with respect to the multiple parallel hits hypothesis, and includes factors such as endotoxin and oxidative stress that are currently perceived as key contributors to the progression from fatty liver to NASH (new reference No. 4: Tilg H et al., Hepatology 2010). We have also recently shown that administration of fructose resulted in an increase in the extent of transport of endogenous gut-derived bacterial endotoxin by the portal vein; this leads to liver fibrosis and hepatocarcinogenesis via induction of lipopolysaccharide (LPS) / Toll-like receptor 4 signaling in a choline-deficient l-amino-acid-defined (CDAA)-fed rat model (new reference No. 5: Seki K et al., Oncotarget 2018). We have revised the manuscript accordingly and we have included this information on page 2, lines 40–53.
- The description in the Results section appears to mention some data not present in the manuscript to describe the rationale between figures. For example, why did the author check the TNF-a, IL6, and CCL2 as proinflammatory cytokines? If these cytokines are already known to be specific to CDAA-induced fibrosis, the authors need to mention this using references. Moreover, the Discussion section is very lengthy and is less relevant to the main notion.
Author response
Thank you for your valuable comments. Miura et al have shown that proinflammatory cytokines and receptors including the C-C motif chemokine receptor 2 (CCR2), tumor necrosis factor-α (TNFα) and interleukin 6 (IL-6) derived from Kupffer cells all contribute to the progression of NASH and to the development of HCC in the CDAA diet-induced NASH model (new reference No. 6: Miura K et al., Gastroenterology 2010; new reference No.7: Miura K et al., Am J Physiol Gastrointest Liver Physiol 2012; new reference No.8: Miura K., J Biol Chem 2016; new reference No. 9: Miura K et al., Hepatology 2013). We have revised the manuscript accordingly and we have included mention of all of these findings on page 2, lines 53–56 and page 10, lines 241–255 and lines 270–286.
- In the figure legends, please include only a description of the experimental design; do not interpret the data. For example, in Figure 4a, it is unclear which assay was used to assess the proliferation, while every figure legend repeats the data interpretation text that is similar to that in the Results section.
Author response
Thank you for pointing this out. We have revised the manuscripts accordingly. We have removed interpretations from the Figure Legends and we have clarified the assays used throughout.

Reviewer 2 Report
The authors investigated the effect of sodium-glucose cotransporter-2 inhibitor (canagliflozin, CA) and dipeptidyl peptidase-4 inhibitor (teneligliptin, TE) in combination on NASH progression in a rat model of steatohepatitis, with choline-deficient L-amino acid-defined (CDAA)-feeding. They found that CA and TE suppressed CDAA diet-induced hepatic fibrogenesis and carcinogenesis. Combined treatment showed stronger inhibitory effects than monotherapy as well as suppression of HSC activation, neovascularization, and oxidative stress. CA alone or combined with TE exhibited significant anti-inflammatory effects. They concluded that that CA and TE have synergistic effects on hepatic fibrogenesis and carcinogenesis with concurrent suppression of HSC and HCC proliferation, angiogenesis, oxidative stress, and inflammation. As known, HCC has had a continuous increase worldwide in incidence over the last two decades, while nonalcoholic steatohepatitis (NASH) is the most severe form of non-alcoholic fatty liver disease (NAFLD) and a potential precursor of HCC. However, therapeutic options for HCC is limited. Therefore, it is important to explore the potential therapeutic strategy of HCC, especially NASH associated HCC development, performing such study is fairly straightforward. Here are several comments:
- General findings in table 1. Because there is no statistically significant differences for the TG levels (210.9 +/-39. Vs 210.4 +/-18.4) and the insulin levels (0.82+/- 0.34 vs 0.86+/- 0.61) between CSAA and CDAA. These data do not support the success of NASH establishment. In addition, NASH histology and NAFLD Active Score (NAS) system are well accepted for diagnosis of NASH. Therefore, need to provide histology and NAS to confirm that NASH has been successfully established in rats by CDAA. Oil-red-o staining is also helpful to show the lipid accumulation in hepatocytes to determine steatosis.
- In vitro effects of CA and TE on Ac-HSCs. It’s unclear how the proliferation assay was determined in the isolated Ac-HSCs, how long the cells were cultured before treatment? The authors should include all the critical experimental details, including isolation of Ac-HSCs, in the method section.
- Effect of CA and TE on EC tube formation in vitro. It’s unclear why EC tube formation could help to determine hepatocarcinogenesis, why the EC tube formation can be used for in vitro angiogenesis? As known, the epithelial–mesenchymal transition (EMT) could be the potential malignant phenotype. A dual staining using the biomarkers of EMT and the biomarkers for angiogenesis such as the VEGF could be more helpful.
- Figure 6. (A) Representative photomicrographs of GST-P-positive hepatic preneoplastic lesions. Need to provide high resolution images to show the cytological features of preneoplastic cells.
- The English issue. It is necessary to have an English-speaking person to further check the language used in this manuscript.
Author Response
Reviewer: 2
Reviewer(s)' Comments to Author:
The authors investigated the combined effect of sodium-glucose cotransporter-2 inhibitor (canagliflozin, CA) and dipeptidyl peptidase-4 inhibitor (teneligliptin, TE) in NASH progression in a rat model of steatohepatitis, with choline-deficient L-amino acid-defined (CDAA)-feeding. They found that CA and TE suppressed CDAA diet-induced hepatic fibrogenesis and carcinogenesis. Compared to monotherapy, combined treatment showed stronger inhibitory effects as well as greater suppression of HSC activation, neovascularization, and oxidative stress. CA alone or VA with TE exhibited significant anti-inflammatory effects. They concluded that CA and TE exert synergistic effects on hepatic fibrogenesis and carcinogenesis with concurrent suppression of HSC and HCC proliferation, angiogenesis, oxidative stress, and inflammation. As known, the global incidence of HCC has increased using the previous two decades, while nonalcoholic steatohepatitis (NASH) is the most severe form of non-alcoholic fatty liver disease (NAFLD) and a potential precursor of HCC. However, therapeutic options for HCC are limited. Therefore, it is crucial to explore the potential therapeutic strategy of HCC, especially NASH-associated HCC development; such a study can be performed easily. Here are several comments:
Major revisions;
- General findings in table 1. There was no significant difference in the TG levels (210.9 ± 39 vs. 210.4 ± 18.4) and insulin levels (0.82 ± 0.34 vs. 0.86 ± 0.61) between CSAA and CDAA. These data do not support the success of NASH establishment. In addition, NASH histology and NAFLD Active Score (NAS) system are well accepted for establishing a diagnosis of NASH. Therefore, there is a need to provide histology and NAS to confirm that NASH has been successfully established in rats with CDAA. Oil-red-o staining is also helpful in determining the lipid accumulation in hepatocytes to diagnose steatosis.
Author Response:
We greatly appreciate your constructive comments. Changes in NAFLD active (NAS) scores are shown in a new Table 2. Microscopic examination revealed significant reductions in steatosis, lobular inflammation and hepatocellular ballooning in groups G3 and G5 (i.e., those treated with CA and CA + TE, respectively) compared to what was observed among the G2 rats (fed a CDAA diet). These changes were accompanied by a significant decrease in the alanine aminotransferase (ALT) level, indicating that CA, but not TE, had both cytoprotective and anti-inflammatory effects on target hepatocytes. We have included a full description of these findings on page 10, lines 217–223.
- The in vitro effects of CA and TE on Ac-HSCs. It is unclear how the proliferation assay was determined in the isolated Ac-HSCs and for how long the cells were cultured before the treatment? The authors should include all the critical experimental details, including isolation of Ac-HSCs, in the Methods section.
Author Response:
Thank you for your valuable comments. After culture for five days, the HSCs showed myofibroblast-like characteristics with reduced lipid vesicles and increased expression of α-smooth muscle actin (SMA). After activation of HSCs by culture on plastic plates for seven days, all cells were distributed homogeneously and were all α-SMA-positive. We have included a description of these findings in the revised manuscript on page 13, lines 355–358.
- From the effect of CA and TE on EC tube formation in vitro, it is unclear why EC tube formation could help determine hepatocarcinogenesis and why the EC tube formation can be used for in vitro angiogenesis. As known, the epithelial–mesenchymal transition (EMT) could be the potential malignant phenotype. A dual staining technique using the biomarkers of EMT and the biomarkers for angiogenesis, such as VEGF could be more helpful.
Author Response: Thank you for your valuable comments. We did not include images or any quantitative analysis of EC tube formation. However, we performed a more extensive examination of the in vitro effects of CA and TE on the expression of E-cadherin and VEGF in primary human umbilical vein endothelial cell (HUVEC) culture. CA and TE, each acting alone, promoted increased expression of the epithelial marker, E- cadherin, and at the same time suppressed the expression of the angiogenesis marker, VEGF, in HUVEC culture (Figs. 8d and 8e). Simultaneous administration of both agents resulted in more potent stimulatory and inhibitory effects on E-cadherin and VEGF expression, respectively, than detected in response to either agent alone. We have included a full description explaining these important findings on page 8, lines 192–199.
- Figure 6. (A) Representative photomicrographs of GST-P-positive hepatic preneoplastic lesions. Please provide high-resolution images to show the cytological features of preneoplastic cells.
Author Response:
Thank you for your valuable comments. The livers of rats on the CDAA diet for 16 weeks exhibit features of fatty cirrhosis, including numerous and various sized neoplastic nodules with aberrant histologic, architectural, and cytoplasmic features together with nuclear atypia (new Fig. 6a and 6b). We have included this description on page 6, line 164–168.
- I would highly recommend that you avail the editing services of an English-speaking expert to ensure correct usage in this manuscript.
Author Response: Thank you for your valuable comments. We had our manuscript reviewed by a native English speaker.

Reviewer 3 Report
In this manuscript, the authors presented that administration of SGLT2-I and DPP4-I suppressed NASH progression in a rat model. They also showed TE and CA inhibited cell proliferation and TGF-b expression in HSCs in vitro study. The authors already reported that SGLT2-I ameliorates liver fibrosis (ref. 8), and that DPP4-I also attenuates liver fibrosis (ref. 13). We can easily speculate combination therapy is a good candidate for treatment of NASH. Thus, these results are not fascinating and not so interesting. We have to think whether this therapy is really effective in clinical setting.
Major
- In the Materials and Methods Section, the authors explained G4 rats were administrated clinically equivalent doses of TE (10mg/kg/day). TE is used at the dosage of 20-40mg/day in clinical setting, so 10mg/kg/day is too much for rats. The authors have to check the serum concentration and confirm that this dosage of TE is appropriate for rats. If not, less dosage of TE should be treated. How about adverse events? Two randomized controlled trials already showed DPP4-I did not attenuate liver fibrosis, and it would be meaningless if excess dose were used.
- Kawaguchi T recently reported that DPP4-I suppressed progression of NASH-related HCC (ref. 15), but in this manuscript, no significant reduction in the proliferation of HCC cell lines in TE-treated group. How do the authors explain this discrepancy?
- There are many types of SGLT2-I and DPP4-I, and CA and TE were used in this experiment. Why did the authors pick up these drugs?
Minor
- The authors misspelled IL-6-6 (line 373, page 14).
Author Response
Reviewer: 3
Comments to the Author
In this manuscript, the authors stated that the administration of SGLT2-I and DPP4-I suppressed NASH progression in a rat model. They also showed TE and CA inhibited cell proliferation and TGF-b expression in HSCs in vitro study. The authors have already reported that SGLT2-I ameliorates liver fibrosis (ref. 8) and that DPP4-I attenuates liver fibrosis (ref. 13). We can easily speculate that the combination therapy is a feasible treatment option for NASH. Thus, these results are not groundbreaking or interesting. We need to examine whether this therapy is really effective in clinical settings.
Comments
1) In the Materials and Methods Section, the authors explained that G4 rats were administrated clinically equivalent doses of TE (10 mg/kg/d). TE is used at a dosage of 20-40 mg/day in the clinical setting; therefore, a dose of 10 mg/kg/d is too high for rats. The authors need to check the serum concentration and confirm that this dosage of TE was appropriate for rats. If not, a lower dosage of TE should be administered. Further, please provide information about adverse events. Two randomized controlled trials have already showed that DPP4-I did not attenuate liver fibrosis, and it would be meaningless if excess doses were used.
Author Response:
Thank you for your thoughtful and constructive comments. As we mentioned in the discussion section (page 2 line 76-79), Japan has just recently approved the use of a fixed-dose combined tablet of teneligliptin (20 mg) and canagliflozin (100 mg). As such, the impact of clinically equivalent doses of CA (100 mg/day) and TE (20 mg/day) and their impact on the progression of NASH progression was the focus of this study. Rats were treated with CA (10mg/kg/day) and TE (0.3mg/kg/day) to approximate clinical dosing strategies. No adverse effects were observed in the present study. I sincerely apologize for the mistake; this has been corrected it in the revised manuscript.
We have included this description on page 11, line 314–315 and footnotes to Table1 and Table2.
2) Kawaguchi T recently reported that DPP4-I suppressed the progression of NASH-related HCC (ref. 15); however, in the present manuscript, no significant reduction was observed in the proliferation of HCC cell lines in the TE-treated group. How do the authors explain this discrepancy?
Author Response:
Thank you for your valuable comments. The drug, sitagliptin has recently been shown to have inhibitory effects on NASH-related progression to HCC via its inhibitory actions at the p62/Keap1/Nrf2-pentose phosphate pathway (new reference No.45: Kawaguchi T et al., Liver Cancer 2019). Furthermore, the dipeptidyl peptidase (DPP)-4 inhibitor, KR62436, promoted primary tumor growth and lung metastasis via induction of the CXCL12/CXCR4/mTOR/EMT signaling axis in a syngeneic mouse model (new reference No.46: Yang F., Cancer Res 2019). The differences in anti-tumor effects of DPP-4 inhibitors might be explained at least in part by differences in pharmacological activities (new reference No. 26: Ceriello A et al., Drugs 2019). We have included a discussion of these findings on page 11, lines 273–279.
3) There are many types of SGLT2-I and DPP4-I; further, CA and TE were used in this experiment. Why did the authors select these drugs?
Author Response:
Thank you for your bringing up this important point. We have recently shown that SGLT2 inhibitors may function as novel anti-cancer agents by inhibiting angiogenesis and progression to HCC in a mouse xenograft mouse model (new reference No. 17: Kaji K., Int J Cancer 2016). As noted above in response to Reviewer #2, the fixed-dose combined tablet of teneligliptin (20 mg) and canagliflozin (100 mg) was recently approved in Japan (new reference No. 26: Ceriello A et al., Drugs 2019) and formed the basis of a clinically meaningful study in this rat model. CA and TE were kindly provided from Mitsubishi Tanabe Pharma Co. Ltd. (Osaka, Japan). We have included a discussion of these findings on page 2, lines 63-65 and 76–79 and page 11, lines 306–307.
Minor revision
- The authors have misspelled IL-6 (line 351, page 12).
Author Response:
We apologize for this error. This has been fixed, see page 14, line 373.

Round 2
Reviewer 1 Report
I agree with the publication of current version of this manuscript. Thanks
Author Response
N/A
Reviewer 2 Report
N/A
Author Response
N/A
Reviewer 3 Report
The authors explained that they made a mistake, and corrected the dose of TE. This is a critical point, and in this experiment, rats were treated with TE via daily oral gavage. The authors should examine the serum concentration of TE. Then they have to at least confirm that they excess the IC50 (1.14nmol/L), which can inhibit the DPP4 activity. Two randomized controlled trials already showed DPP4-I did not attenuate liver fibrosis, so they should focus on the difference between the randomized trials and this experiment. If the difference is shown, new trials with DPP4-I would be planned in the future.
Author Response
March 9th, 2020
Dear Editor-in-Chief, Associate Editor, and Reviewers:
Please find enclosed our edited manuscript in Word format [file name: ID ijms-729810.R2].
Title: Combined treatment with sodium-glucose cotransporter-2 inhibitor (canagliflozin) and dipeptidyl peptidase-4 inhibitor (teneligliptin) alleviates NASH progression in a non-diabetic rat model of steatohepatitis
Authors: Takahiro Ozutsumi, Tadashi Namisaki, Naotaka Shimozato, Kosuke Kaji, Yuki Tsuji, Daisuke Kaya, Yukihisa Fujinaga, Masanori Furukawa, Keisuke Nakanishi, Shinya Sato, Yasuhiko Sawada, Soichiro Saikawa, Kou Kitagawa, Hiroaki Takaya, Hideto Kawaratani, Kei Moriya, Takemi Akahane, Akira Mitoro, Hitoshi Yoshiji
Name of Journal: International Journal of Molecular Sciences
Manuscript No: ijms-729810. R2
We are very grateful to all of you for providing thoughtful comments and suggestions regarding our manuscript. We have addressed all concerns and issues and have revised our manuscript accordingly, as detailed in the point-by-point response below. All changes in the revised version are highlighted in blue. We believe that the manuscript has been greatly improved and we hope it has reached the standards for publication in International Journal of Molecular Sciences. Once again, we acknowledge your comments which have been extremely valuable in improving the quality of our manuscript.
Reviewer: 3
Reviewer(s)' Comments to Author:
The authors explained that they made a mistake, and corrected the dose of TE. This is a critical point, and in this experiment, rats were treated with TE via daily oral gavage. The authors should examine the serum concentration of TE. Then they have to at least confirm that they excess the IC50 (1.14nmol/L), which can inhibit the DPP4 activity. Two randomized controlled trials already showed DPP4-I did not attenuate liver fibrosis, so they should focus on the difference between the randomized trials and this experiment. If the difference is shown, new trials with DPP4-I would be planned in the future.
Authors response
Thank you for your constructive comments. We would like apologize for not responding to your previous comment. The serum concentrations of TE were determined by liquid chromatography–tandem mass spectrometry1,2. Analysis was performed using the LCMS-8060 triple quadrupole mass spectrometer with Nexera UHPLC system (Shimadzu Corporation, Kyoto, Japan)3. The concentrations of TE were 0.83 ± 0.12 and 0.89 ± 0.11 nmol/L in G4 and G5 (which were treated with TE and CA + TE, respectively), which were markedly lower than the IC50 value of 18 nmol/L for sitagliptin4. Plasma DPP-4 activity was measured using the DPP-4 activity assay kit (Biovision, Milpitas, CA, USA)5. DPP-4 activity was significantly augmented in G2 (CDAA group) than in G1 (CSAA group) and was significantly suppressed in G4 and G5 than in G2 (supplementary Fig.1, for reviewer only)
References
- Eto T, Inoue S, Kadowaki T. Effects of once-daily teneligliptin on 24-h blood glucose control and safety in Japanese patients with type 2 diabetes mellitus: a 4-week, randomized, double-blind, placebo-controlled trial. Diabetes Obes Metab 2012;14:1040-1046.
- Kadowaki T, Kondo K. Efficacy, safety and dose-response relationship of teneligliptin, a dipeptidyl peptidase-4 inhibitor, in Japanese patients with type 2 diabetes mellitus. Diabetes Obes Metab 2013;15:810-818.
- Rush MD, van Breemen RB. Role of ammonium in the ionization of phosphatidylcholines during electrospray mass spectrometry. Rapid Commun Mass Spectrom 2017;31:264-268.
- Bergman A, Ebel D, Liu F, Stone J, Wang A, Zeng W, Chen L, et al. Absolute bioavailability of sitagliptin, an oral dipeptidyl peptidase-4 inhibitor, in healthy volunteers. Biopharm Drug Dispos 2007;28:315-322.
- Kim SJ, Kwon SK, Kim HY, Kim SM, Bae JW, Choi JK. DPP-4 inhibition enhanced renal tubular and myocardial GLP-1 receptor expression decreased in CKD with myocardial infarction. BMC Nephrol 2019;20:75.
Round 3
Reviewer 3 Report
The authors have to explain why TE worked well though the concentration of TE was lower than IC50. Higher concentration of TE would be more effective. If these results are true, clinical trials with higher dosage of TE should be examined again.
This is an important point, and these results and explanations should be inserted to the text in this manuscript.
Author Response
March 19, 2020
Dear Editor-in-Chief, Associate Editor, and Reviewers:
Please find enclosed our edited manuscript in Word format [file name: ID ijms-729810.R3].
Title: Combined treatment with sodium-glucose cotransporter-2 inhibitor (canagliflozin) and dipeptidyl peptidase-4 inhibitor (teneligliptin) alleviates NASH progression in a non-diabetic rat model of steatohepatitis
Authors: Takahiro Ozutsumi, Tadashi Namisaki, Naotaka Shimozato, Kosuke Kaji, Yuki Tsuji, Daisuke Kaya, Yukihisa Fujinaga, Masanori Furukawa, Keisuke Nakanishi, Shinya Sato, Yasuhiko Sawada, Soichiro Saikawa, Koh Kitagawa, Hiroaki Takaya, Hideto Kawaratani, Kei Moriya, Ryuichi Noguchi, Takemi Akahane, Akira Mitoro, Hitoshi Yoshiji
Name of Journal: International Journal of Molecular Sciences
Manuscript No: ijms-729810. R3
We are very grateful to all of you for providing thoughtful comments and suggestions regarding our manuscript. We have addressed all concerns and issues and have revised our manuscript accordingly, as detailed in the point-by-point response below. All changes in the revised version are highlighted in blue. We believe that the manuscript has been greatly improved and we hope it has reached the standards for publication in International Journal of Molecular Sciences. Once again, we acknowledge your comments which have been extremely valuable in improving the quality of our manuscript.
Reviewer: 3
Reviewer(s)' Comments to Author:
The authors have to explain why TE worked well though the concentration of TE was lower than IC50. Higher concentration of TE would be more effective. If these results are true, clinical trials with higher dosage of TE should be examined again.
This is an important point, and these results and explanations should be inserted to the text in this manuscript.
Authors’ response
Thank you for your constructive comments. I sincerely apologize for this major misunderstanding. Serum TE concentration was determined by liquid chromatography–tandem mass spectrometry (new reference No. 49: Eto T et al., Diabetes Obes Metab 2012 and new reference No. 50: Kadowaki T et al., Diabetes Obes Metab 2013). Reanalysis was performed using the LCMS-8060 triple quadrupole mass spectrometer with the Nexera UHPLC system (Shimadzu Corporation, Kyoto, Japan) (new reference No. 51: Rush MD et al., Rapid Commun Mass Spectrom 2017). Serum TE concentration was confirmed to be 1.73 ± 0.27 and 1.88 ± 0.31 nmol/L in G4 and G5 (which were treated with TE and CA + TE, respectively); these values were higher than the IC50 value of TE (1.14 nmol/L). Further, plasma DPP4 activity was measured using a DPP4 activity assay kit (Biovision, Milpitas, CA, USA) (new reference No. 52: Kim SJ et al., BMC Nephrol 2019). DPP4 activity was significantly augmented in G2 (CDAA group) compared with in G1 (CSAA group) and was significantly suppressed in G4 and G5 compared with in G2 (supplementary Fig. 1, for reviewer only). We have included a detailed description explaining these important findings on page 10, lines 241–251 and page 12, lines 340–344.
References
- Eto T, Inoue S, Kadowaki T. Effects of once-daily teneligliptin on 24-h blood glucose control and safety in Japanese patients with type 2 diabetes mellitus: a 4-week, randomized, double-blind, placebo-controlled trial. Diabetes Obes Metab 2012;14:1040-1046.
- Kadowaki T, Kondo K. Efficacy, safety and dose-response relationship of teneligliptin, a dipeptidyl peptidase-4 inhibitor, in Japanese patients with type 2 diabetes mellitus. Diabetes Obes Metab 2013;15:810-818.
- Rush MD, van Breemen RB. Role of ammonium in the ionization of phosphatidylcholines during electrospray mass spectrometry. Rapid Commun Mass Spectrom 2017;31:264-268.
- Kim SJ, Kwon SK, Kim HY, Kim SM, Bae JW, Choi JK. DPP-4 inhibition enhanced renal tubular and myocardial GLP-1 receptor expression decreased in CKD with myocardial infarction. BMC Nephrol 2019;20:75.